# Resolution of *RHCE* Haplotype Ambiguities in Transfusion Settings

**DOI:** 10.3390/ijms25115868

**Published:** 2024-05-28

**Authors:** Caroline Izard, Laurine Laget, Sophie Beley, Nelly Bichel, Lugdivine De Boisgrollier, Christophe Picard, Jacques Chiaroni, Julie Di Cristofaro

**Affiliations:** 1Etablissement Français du Sang PACA Corse, 13005 Marseille, France; 2Aix Marseille University CNRS EFS ADES UMR7268, 13015 Marseille, France

**Keywords:** red blood cell transfusion, *RHCE* haplotype, genetic polymorphisms, mRNA, reticulocyte, sequencing

## Abstract

Red blood cell (RBC) transfusion, limited by patient alloimmunization, demands accurate blood group typing. The Rh system requires specific attention due to the limitations of serological phenotyping methods. Although these have been compensated for by molecular biology solutions, some RhCE ambiguities remain unresolved. The *RHCE* mRNA length is compatible with full-length analysis and haplotype discrimination, but the *RHCE* mRNA analyses reported so far are based on reticulocyte isolation and molecular biology protocols that are fastidious to implement in a routine context. We aim to present the most efficient reticulocyte isolation method, combined with an RT-PCR sequencing protocol that embraces the phasing of all haplotype configurations and identification of any allele. Two protocols were tested for reticulocyte isolation based either on their size/density properties or on their specific antigenicity. We show that the reticulocyte sorting method by antigen specificity from EDTA blood samples collected up to 48 h before processing is the most efficient and that the combination of an *RHCE*-specific RT-PCR followed by *RHCE* allele-specific sequencing enables analysis of cDNA *RHCE* haplotypes. All samples analyzed show full concordance between *RHCE* phenotype and haplotype sequencing. Two samples from the immunohematology laboratory with ambiguous results were successfully analyzed and resolved, one of them displaying a novel *RHCE* allele (*RHCE**03 c.340C>T).

## 1. Introduction

Red blood cell (RBC) transfusion is an important therapeutic option for various hematologic disorders, such as sickle cell disease (SCD), autoimmune hemolytic anemia, thalassemia, myelodysplastic syndrome, or hematopoietic stem cell transplantation [1]. RBC transfusion is, however, limited by the availability of compatible transfusion units and its counterpart, patient alloimmunization, potentially leading to acute or Delayed Hemolytic Transfusion Reactions (DHTR) [2,3]. 

To ensure recipient/donor compatibility and prevent alloimmunization against RBC antigens and subsequent adverse events, accurate blood group typing is mandatory [4]. The Rh system requires specific attention and expertise because of (i) its highly immunogenic properties, (ii) limitation of serological phenotyping methods, and (iii) broad genetic diversity, especially in people of African descent [5,6,7,8,9].

The Rh system, coding for transmembrane RhD and RhCE proteins with an undisclosed function, is expressed on the RBC surface as a complex with the Rh-associated glycoprotein. RhD and RhCE polypeptides are produced from adjacent homologous genes, *RHD*, and *RHCE*, located on chromosome 1, the former having arisen by duplication of the second during primate evolution [10]. The Rh system comprises 56 antigens, encoded by either the *RHD* gene, the *RHCE* gene, or both. RhD antigens are the D antigen (RH1) and its many variants that result in weakened or partial expression (about 200 variants)—the d phenotype refers to the deletion or absence of a functional *RHD* gene [11]. The main antigens of the RhCE protein are the C (RH2), E (RH3), c (RH4), and e (RH5) antigens and over 115 variants associated with one of the four *RHCE* alleles: *RHCE*ce* (*RHCE***01*), *RHCE*Ce* (*RHCE***02*), *RHCE*cE* (*RHCE*03*), or *RHCE*CE* (*RHCE*04*) [11]. Although not all of the allelic variants have an impact on the phenotype and are of clinical interest, accurate characterization of the most frequent variants or of those implicated in transfusion settings is mandatory in clinical practice. 

In the context of routine immunohematology, serological techniques enable us to characterize most of the Rh phenotypes, and within the last decade, molecular biology solutions have helped compensate for the main limitations of serological techniques [12,13,14,15,16,17,18]. Still, in some situations, ambiguities or discrepancies cannot be resolved, especially for RhCE, and uncertainties remain regarding the transfusion instructions; in this context, the French Blood Center, whose main mission is to ensure France’s self-sufficiency in blood products under optimal safety and quality conditions, is continuously setting up molecular typing strategies in line with technological developments [18,19,20,21,22,23,24,25,26,27,28,29].

For the identification of combined antigens RH6 (ce), RH7 (Ce), RH22 (CE), or RH27 (cE), for which no serological reagent is available, a simple method based on molecular biology is lacking. Indeed, no solution based on DNA analysis can offer haplotype phasing, notably because of the structure of the *RHCE* gene, spanning over nearly 60 Kilobases (Kb) with non-polymorphic introns and whose single-nucleotide polymorphisms (SNPs) responsible for the C/c and E/e antigens (respectively, rs676785 in exon 2 and rs609320 in exon 5) are about 18 Kb apart. Consequently, accurate RhCE typing is complicated for individuals with a heterozygous *RHCE* genotype and presenting an SNP that cannot be phased with one of the two different *RHCE* haplotypes. In such situations, putative haplotypes may only be deduced from the serological result, variants described in the literature and databases, their reported frequency, and patient geographic origin. 

From our experience, the most reported RhCE typing ambiguities in a clinical context are ceSL/CE (*RHCE*01.10/*04*) or CeJAHK/cE (*RHCE*02.03/*03*); Ce(667)/ce(W16C) (*RHCE*02.22/*01.01*) or Ce/ceMO (*RHCE*02/*01.07*); CE/ceMo (*RHCE*04/*01.07*) or Ce(667)/cE (*RHCE*02.22/*03*); ceHAR/Ce (*RHCE*01.22/*02*) or CeVA/ce (*RHCE*02.04/*01*); and ceMo/CE (*RHCE*01.07/*04*) or Ce(667)/cE (*RHCE*02.22/*03*) [4,11,30,31,32].

Unlike the *RHCE* gene, its 1254 base-long mRNA transcript is compatible with full-length analysis and haplotype discrimination. Accordingly, studies on the Rh system focused on mRNA to improve genotype results [33,34,35,36,37]. However, due to a high level of homology between *RHCE* and *RHD* mRNA, molecular biology protocols are often incremental and complex.

Moreover, *RHCE* and *RHD* mRNA expression is restricted to reticulocytes, which represent 0.2% to 2.0% of peripheral blood cells [37,38,39]. Based on these physico-biological characteristics, different methods were set up to isolate reticulocytes from peripheral blood, such as density gradient centrifugation [40]. Reticulocyte isolation is also possible because they differ from erythrocytes by their transferrin receptor (CD71) expression, which disappears during maturation and constitutes a reticulocyte-specific surface protein [41].

*RHCE* mRNA analysis requires reticulocyte isolation and multifarious molecular biology protocols that are cumbersome to implement in a routine context. 

We aimed to compare the efficiency of these reticulocyte isolation methods based either on their size/density properties or on their specific antigenicity, and to validate the most straightforward molecular biology method to analyze cDNA *RHCE* haplotypes.

We present, in this study, a combined protocol of reticulocyte isolation based on positive selection with *RHCE* allele-specific analysis to resolve RhCE ambiguous typing results in the context of routine immunohematology.

## 2. Results

### 2.1. Reticulocyte Antigen-Specific Isolation Shows Sufficient RNA Quantity Retrieval

A comparison of the two protocols applied for reticulocyte isolation shows that more RNA, estimated by *18S* and *GAPDH* Q-PCR analysis, was recovered from reticulocytes isolated by antigen specificity, as compared to size/density properties-based methods (Figure 1). Analysis by flow cytometry assesses the purity of reticulocytes (Appendix A). 

Considering the number of columns used for the reticulocyte antigen-specific isolation regarding the quantity of RNA retrieved, the use of three columns displayed a better yield, considering experiment duration vs. real-time PCR results compared to one, four, or six columns per sample (Figure 1). Indeed, no difference was observed in *18S* ribosomal RNA real-time PCR results between three, four, or six columns and the control.

RNA recovery according to the delay between blood sample collection in EDTA tubes and processing at 24, 36, 48 h, and 7 days showed equivalent results at 24 and 48 h using three columns (Appendix A). 

### 2.2. RHCE Haplotypes Are Distinguished by RHCE-Specific RT-PCR and Monoallelic Sequencing

The association of *RHCE*-specific reverse transcription (RT) with *RHCE*-specific amplification shows that the *RHCE* sequence was retrieved without *RHD* contamination (Appendix A and Appendix A). 

*RHCE**c and *RHCE***C* allele-specific sequencing enables independent cDNA *RHCE* haplotype analysis, as a hemizygous signal at each heterozygous position was obtained using each sequencing primer (Appendix A).

The analysis of the twelve samples used to validate the protocol from reticulocyte isolation to *RHCE* mRNA analysis revealed a full concordance of RhCE phenotype and *RHCE* haplotype sequencing (Table 1). 

The analysis of the 30 RNA samples extracted from peripheral blood collected in PAXgene^®^ Blood RNA tubes used to validate the *RHCE* mRNA protocol also revealed a full concordance of RhCE phenotype and *RHCE* haplotype sequencing (Appendix A). 

### 2.3. Resolution of RHCE Haplotype Ambiguities and Novel Haplotype Identification

The *RHCE*-specific RT-PCR followed by the *RHCE**c and *RHCE***C* allele-specific sequencing protocol was applied to two samples with ambiguous results from the immunohematology laboratory in a routine context; both samples were successfully analyzed and resolved. 

The first case was resolved and displayed an *RHCE***02/RHCE*03.13* genotype, i.e., displaying a c.728A>G polymorphism in the 5th exon associated with the *RHCE*cE* allele (*RHCE**03.13) (Appendix A). Therefore, the phenotype was confirmed as C, Ew, cw, and e, and the transfusion order was RH −3, −4.

The second case showing incoherence between the observed phenotype (RH: −3, 4) and the possible genotypes deduced from Bioarray results (*RHCE***01.20.07* (RHCE*ceJAL)/*RHCE*04* or *RHCE*02.01* (RHCE*CeMA)/*RHCE***03*) was resolved thanks to the *RHCE* cDNA analysis of the patient’s offspring. The *RHCE* cDNA haplotype analysis revealed a novel *RHCE* allele displaying a c.340T polymorphism (rs148487630) in the 3rd exon associated with the *RHCE*cE* allele (*RHCE***03* c.340C>T) (Appendix A) responsible for an amino acid substitution, arginine to tryptophan (Arg>Trp), at position 114 predicted on the external loop 2 of the polypeptide. This new allele coding sequence is given in Appendix A (Genbank Accession Number PP583668). The second *RHCE* allele of the offspring was *RHCE*02*, and the deduced genotype of the mother was *RHCE*cE340T/*02*. Thus, for both mother and offspring, considering their C/e phenotype, the transfusion order was RH −3, −4. 

## 3. Discussion

In this study, we focused on the *RHCE* gene, which is one of the most clinically relevant blood group systems in transfusion as it displays high immunogenicity and great diversity [2,3,4,5,6,7,8,9]. In routine immunohematology, the drawbacks of serological techniques are compensated for by genotyping methods [12,13,14,15,16,17,18]. In some situations, however, samples may show ambiguities or discrepancies that require custom analyses. 

In this study, we present a protocol to accurately identify and discriminate between *RHCE* haplotypes. To circumvent the *RHCE* gene size that impedes full-length sequencing from genomic DNA, we chose to analyze *RHCE* mRNA. As *RHCE* mRNA expression is restricted to reticulocytes [37,38,39], we compared the efficiency of two methods of reticulocyte isolation.

Reticulocyte isolation was performed either by CD71-positive selection by immunomagnetic cell separation or by reticulocyte size and density properties, combining cellulose fiber-based chromatography and gradient density centrifugation [40]. Q-PCR analysis showed that positive selection by immunomagnetic separation enabled the recovery of a larger amount of RNA, which facilitates subsequent molecular biology analyses. Furthermore, this protocol can be performed faster and is less complicated to implement. Our results show that the best performance regarding reagent and time consumption was obtained using three separation columns per blood sample. No cytometry or molecular biology analysis was carried out to check for leukocyte or platelet contamination.

Following RNA extraction, a three-step RT-PCR-sequencing specific to *RHCE* haplotypes was set up to phase genetic polymorphism(s) to each *RHCE* allele. Whereas most protocols so far have required several PCRs to cover all *RHCE* polymorphisms [34,35,36,37], our protocol allows for analysis from exon 2 to exon 6 of the mRNA. 

Two ambiguous samples from the immunohematology laboratory were successfully resolved. Concerning the first case, an *RHCE***02/RHCE*03.13* genotype was confirmed, i.e., displaying a c.728A>G polymorphism in the 5th exon associated with an *RHCE***03* sequence (*RHCE**03.13). For the second case, displaying incoherence between the genotype and phenotype, a novel allele was identified (*RHCE***03* c.340C>T), and, accordingly, ambiguity was resolved. The new *RHCE* allele displays a c.340T polymorphism (rs148487630) in the 3rd exon, which has been described in an *RHCE*02.01* (*RHCE*Ce.01*; *RHCE*CeMA; RHCE*CeJAL*) allele [33] from a French donor with a D+, Cw, E−, c, and e phenotype and associated with a decrease in C-reactivity. Most interestingly, the same mutation associated with an *RHCE***03* allele led to a decrease in C-reactivity. 

Our study supports that the protocol presented here is implementable in immunohematology laboratories and is suitable to solve ambiguities unresolved by serological and basic molecular biology methods in a routine context. Our protocol presents the advantage of covering all configurations as it allows for full-length sequencing of the *RHCE* coding sequence and phasing of each allele. However, it does require a freshly collected blood sample (<48 h), which makes it necessary to ask for a second blood collection once the ambiguity or discrepancy has been observed using routine techniques. We chose to work with fresh EDTA collection tubes, instead of other collection tubes that specifically preserve RNA, to stick to a routine and clinical environment. Reception of a blood sample within 48 h is compatible with the cooperation and sample transport of French immunohematology laboratories and clinical departments.

This solution may yet evolve thanks to technological developments in molecular biology, notably droplet digital PCR (ddPCR), which can also be used to phase variants [42,43]. ddPCR enables PCR amplification from a single DNA molecule in a nanoliter-sized droplet and can be combined with allele-specific TaqMan^®^ probes, to rapidly phase two SNP loci thousands of bases apart. Accordingly, this method would be suitable to phase the described polymorphisms and, notably, SNPs responsible for the C/c and E/e antigens (about 18 Kb apart). In this intended use, such a method would have the advantage of being applicable from DNA, avoiding reticulocyte isolation, RNA extraction, and the RT step. Targeted ddPCR would also be applicable to solve the most reported RhCE typing ambiguities in routine contexts by targeting specific SNPs, such as c.48G>C, c.365C>T, and c.667G>T, allowing for discrimination between ceSL (*RHCE*01.10*), CeJAHK (*RHCE*02.03*), Ce(667) (*RHCE*02.22*), ce(W16C) (*RHCE*01.01*), and ceMO (*RHCE*01.07*) alleles [11]. More efforts would be necessary, however, to characterize alleles, such as CeVA (*RHCE*02.04*) or ceHAR (*RHCE*01.22*), that are defined by eight polymorphisms [11]. In addition, defining the many existing *RHCE* variants thanks to this method may result in an unbalanced cost–benefit ratio. For these, and for unknown polymorphisms, i.e., novel alleles, Sanger sequencing still remains the gold-standard method. 

It should be stressed that analysis of the Rh system requires a deep knowledge of the existing variants and the relationship between the homologous *RHD* and *RHCE* genes, whatever the methodological developments and progress made in that field. In some situations, Rh phenotype identification requires analysis of both genes; for example, a Cw phenotype, in the absence of the 109pb insertion in exon 2 of *RHCE*, may be due to an *RHD***DAU5* allele [19]; an RhD phenotype, in the absence of the *RHD* gene, may be due to an *RHCE*ceHAR* allele [44]; identification of an RH:-18 phenotype in an *RHD*DAR* context requires definition of the *RHCE* genotype, i.e., *RHCE*ceAR/RHCE*ce* or *RHCE*ceAR*/*RHCE*ceAR*. 

In conclusion, accurate identification of the RhD and RhCE phenotype remains a complex topic that calls for complementary methodologies, expertise in genetics, and experience in immunohematology.

## 4. Materials and Methods

### 4.1. Samples

Peripheral blood was obtained from healthy donors or patients after obtaining written informed consent. The donations were collected in accordance with French blood donation regulations and ethics and the French Public Health Code (Article L.1221-1). Peripheral blood samples were collected in EDTA tubes for both reticulocyte isolation and subsequent mRNA *RHCE* analysis, RhCE phenotyping, and *RHCE* genotyping. 

RNA isolated from peripheral blood collected in PAXgene^®^ Blood RNA tubes from sickle cell disease patients enrolled at the Internal Medicine Department, AP-HM, Marseille, after obtaining written informed consent (protocol approved by the Ethics Committee Sud Mediterranée IV; ID-RCB: 2017-A02744-49) with known *RHCE* genotype and RhCE phenotype was used as controls in molecular biology experiments. 

Immortalized human erythroblast cell line BEL-A was used as a positive control in molecular biology experiments [45].

### 4.2. RHCE Phenotyping and Genotyping

RhCE phenotyping was performed according to a three-line technique. First-line RhCE phenotyping was a microplate technique (Qwalys EVO; DIAGAST, Loos, France), the second-line technique was performed using a microfiltration method (OCD Auto-Vue Innova; Ortho Clinical Diagnostics, Raritan, NJ, USA), and the third-line technique used saline-filled test tubes with monoclonal antibodies (DIAGAST, Loos, France). All methods were performed in accordance with the manufacturer’s recommendations and as described in [19].

DNA was extracted using the automated QIAcube from Qiagen, France, according to the manufacturer’s recommendations. *RHCE* genotyping was performed using the *RHCE* BeadChip array (Bioarray/Immucor) containing 75 markers associated with *RHCE* alleles.

### 4.3. RNA Extraction from Reticulocytes

Two different protocols were applied for reticulocyte isolation, based either on their size/density properties or on their specific antigenicity. Blood samples collected in EDTA tubes were processed within 24 h after collection.

#### 4.3.1. Reticulocyte Sorting According to Size and Density

Reticulocytes were isolated according to a published protocol [40], combining cellulose fiber-based chromatography and gradient density centrifugation. Briefly, 4 milliliters (mL) of blood sample was pelleted by centrifugation, leukocytes were removed through cellulose fibers (Sigma-Aldrich, Saint-Quentin-Fallavier, France), and the diluted pass-through material was separated by centrifugation through 62% and 72.5% Percoll layers (VWR-Avantor, Rosny-sous-Bois, France). After centrifugation, the cell pellets were used for subsequent analysis. Two blood samples were analyzed in duplicate using this protocol.

#### 4.3.2. Reticulocyte Antigen-Specific Isolation 

Reticulocytes were isolated from the blood sample thanks to their CD71 antigen-specific expression (transferrin receptor) using CD71 magnetic MicroBeads and MiniMACS Separation Columns, type LS (Miltenyi, Paris, France), according to the manufacturer’s recommendations: 600 µL of peripheral blood supplemented with buffer solution (MACS BSA Solution diluted with autoMACS Rinsing Solution, Miltenyi Biotec) in a final volume of 2 mL was centrifuged (300× *g*, 10 min). Pellets suspended in the buffer solution were incubated with 30 µL of microbeads for 15 min at 4 °C, washed (300 g, 10 min), and resuspended in a final volume of 2 mL buffer solution. Isolation was performed on LS MiniMACS Separation Columns placed in a magnetic field with 3 washing steps, beads were collected in 5 mL of buffer solution, centrifuged (300× *g*, 10 min), and resuspended in 200 µL of buffer solution before subsequent analysis. 

To estimate the protocol’s yield according to the number of columns, reticulocytes were isolated using one, three, four, and six columns; ten blood samples were analyzed in duplicate using this protocol (Table 1).

Based on the best yield according to the number of columns (i.e., three columns, see Results section), RNA retrieval was analyzed according to the time between blood sample collection in EDTA tubes and processing. Six blood samples collected in EDTA tubes were processed at 24, 36, and 48 h, and 7 days after collection.

#### 4.3.3. Reticulocyte Isolation Assessment by Cytometry 

Reticulocytes isolated from blood samples were stained for flow cytometry analysis with the murine anti-CD71 IgG2a antibody clone CY1G4-FITC (Biolegends, Amsterdam, The Netherlands). The murine IgG2a antibody clone S43.10-FITC (Miltenyi, Paris, France) served as the isotype control. Reticulocyte cytometry data were acquired on a Cytoflex machine (Beckman Coulter, Villepinte, France) and analyzed with FlowJo 10 software (BDbiosciences, Rungis, France). The cell population was detected according to its morphology using an initial gate set in an FSC-A/SSC-A plot. Single cells were further gated in an FSC-A/FSC-H plot; finally, cells expressing CD71, as compared to isotype staining, were gated in a subsequent FITC-A/SSC-A plot. 

#### 4.3.4. RNA TRIzol Extraction

RNA was extracted from reticulocyte pellets with TRIzol RNA Isolation Reagents (Invitrogen ThermoFischer, Illkirch-Graffenstaden, France), according to the manufacturer’s recommendations. Briefly, 750 µL of TRIzol was used for cell pellet disruption, 150 µL of chloroform allowed for aqueous phase retrieval, and RNA was precipitated using 375 µL of isopropanol. RNA was washed with ethanol (75%), suspended in 20 µL of RNAse-free water, and its concentration was estimated with Nanodrop^®^ (Thermo Scientific) before further analysis.

#### 4.3.5. Assessment of RNA Extraction Efficiency

Real-time PCR analysis was used to estimate the efficiency of erythrocyte RNA isolation depending on the reticulocyte isolation protocols (i.e., size/density or antigenic specificity, with different numbers of columns) and depending on the time from blood collection to processing.

Total RNA obtained from the different reticulocyte isolation protocols was quantified by real-time PCR analyses following reverse transcription using random hexamers and Superscript IV reverse transcriptase (Invitrogen ThermoFischer, Illkirch-Graffenstaden, France). *18S* ribosomal RNA and *ACTB* (actinβ) mRNA analyses were performed using TaqMan technology (Invitrogen) according to manufacturer’s protocol (respectively, TaqMan assays Hs99999901_s1 and ACTB Hs99999903_m1, Invitrogen). Each experiment was carried out in duplicate using QwantStudio 3 (Invitrogen). The average Ct was calculated with StepOne 2.1 software (Invitrogen), excluding Ct duplicates with a standard deviation above 0.5.

### 4.4. RHCE cDNA Haplotype Analysis

#### 4.4.1. RHCE-Specific Reverse Transcription and RHCE-Specific Amplification

mRNA from the *RHCE* gene was specifically reverse transcribed using 100 ng of total RNA, an *RHCE*-specific primer (final concentration 0.1 µM; Appendix A), and the Superscript IV Reverse Transcriptase (Invitrogen) according to the manufacturer’s protocol.

Specific amplification of *RHCE* cDNA was then performed by PCR using 3 µL of cDNA, the Taq DNA Polymerase recombinant kit (Invitrogen) following the manufacturer’s instruction, and an *RHD*/*RHCE*-specific forward primer and the *RHCE*-specific reverse primer used for the reverse-transcription step (Appendix A, Tm: 60 °C, PCR product length: 995 base pairs). 

#### 4.4.2. RHCE Specificity Assessment of the RT-PCR Protocol

*RHCE* cDNA biallelic PCR products were sequenced by Sanger sequencing (Eurofins Genomics, Ebersberg, Germany) using an *RHCE*-specific sequencing primer (Appendix A). *RHCE*-specific amplification was assessed by analyzing polymorphisms discriminating between *RHCE* and *RHD* sequences (respective GenBank accession numbers NM_020485.8 and NM_016124.6) (Table 2).

The consistency of the biallelic sequencing results was checked by comparing *RHCE* polymorphisms (Table 3 and position c. 676 G>C (rs609320)) with the RhCE phenotype. 

All sequencing analyses were performed by two independent operators using different sequence alignment software (Codon Code Aligner version 10.0.2 (Codon Code Corporation, Centerville, MA, USA) and Lasergene V. 17.5 (DNASTAR Inc., Madison, WI, USA)).

#### 4.4.3. RHCE Monoallelic Sequencing

PCR products were then sequenced using *RHCE**c and *RHCE***C* allele-specific sequencing primers (Appendix A) by Sanger sequencing (Eurofins Genomics, Ebersberg, Germany). *RHCE**c and *RHCE***C* haplotype discrimination was checked at position c.307 (rs676785) (Table 3). 

Then, all *RHCE* polymorphisms were analyzed and assigned to each haplotype: c.676G>C for e/E antigen and polymorphisms of *RHCE* variants from position c.250 in exon 2 to position c.939 at the end of the sixth exon.

### 4.5. Resolution of RHCE Haplotype Ambiguities and Novel Haplotype Identification

The *RHCE* cDNA haplotype protocol under study was used to analyze two samples from our immunohematology laboratory, unresolved by routine methods. 

The first case was a 54-year-old female blood donor displaying serological ambiguities for the RH3 (E) and RH4 (c) antigens. The observed phenotype was RH:1, 2, 3w, 4w, 5 (weakening of RH3 and RH4 antigens); however, depending on the techniques used, the RH3 antigen (E) was negative or weakly positive, while the RH4 antigen (c) was positive with differences in intensity. To resolve these ambiguous serological results, the *RHCE* genetic polymorphisms were analyzed using the Bioarray technique and *RHCE* gene sequencing (as previously described [20]) and were identified as follows: c.48G>C, c.676G>C, c.728A>G and the insertion of 109pb in exon 2 at heterozygous state. According to the ISBT database variants [11], the most likely *RHCE* genotype was *RHCE*02/RHCE*03.13* [3].

The second case was a 33-year-old woman with an RH:1, 2, -3, 4w, 5 phenotype (weakening of RH4 antigen determined using the first-line technique (2+) but not confirmed with the second-line technique (4+)). The *RHCE* gene polymorphisms analyzed using the Bioarray technique were c.48G>C, c.340C>T, c.676G>C, and c.733C>G in the heterozygous state, as well as the insertion of 109pb in exon 2. These results were confirmed by *RHCE* gene sequencing [3]. According to the ISBT database [11], two possible *RHCE* genotypes were deduced: either *RHCE*01.20.07* (RHCE*ceJAL)/*RHCE***04*, or *RHCE*02.01* (RHCE*CeMA)/*RHCE*03*. However, because of the weakening of the RH4 antigen and the absence of the RH3 antigen, no combination was plausible, as both genotypes are discordant with the weakened RH4 antigen and absence of RH3 antigen. Further investigation was, therefore, performed on her child followed-up for polytransfused thalassemia. The child displayed an RH:1,2,-3,4,5 phenotype, which should be considered with caution because of serological technique limitations in polytransfusion contexts and the same *RHCE* genetic polymorphisms as the mother, giving no further clue for the maternal inherited haplotype.

## Figures and Tables

**Figure 1 ijms-25-05868-f001:**
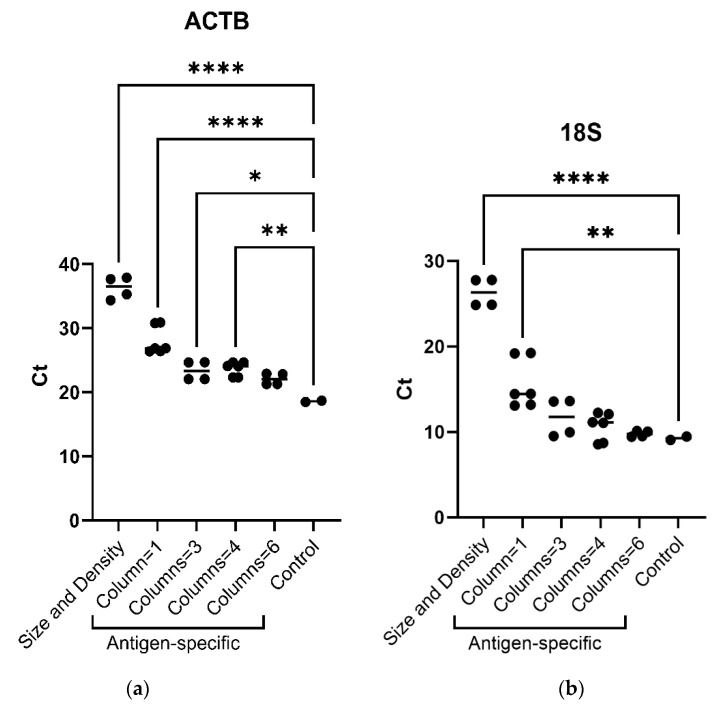
(**a**) *ACTB* (*actinβ*) mRNA and (**b**) *18S* ribosomal RNA real-time PCR analysis (absolute Ct) according to reticulocyte sorting protocols (i.e., size/density and antigenic specificity, using one, three, four or six separation columns). Ordinary one-way ANOVA, multiple comparisons. One asterisk (*) indicates *p* value below 0.05 (*p* < 0.05); two asterisks (**) indicate *p* value below 0.01 (*p* < 0.01); four asterisks (****) indicate *p* value below 0.0001 (*p* < 0.0001)).

**Table 1 ijms-25-05868-t001:** Description of samples used to validate the protocol from reticulocyte isolation to *RHCE* mRNA analysis. RhD and RhC phenotypes, i.e., presence (+) or absence (−) of Rh antigens (RH1 (D), RH2 (C), RH3 (E), RH4 (c), RH5 (e)), and *RHCE* haplotypes are given. Reticulocyte sorting method is shown with the number of columns used per sample concerning the antigen specificity protocol (NA: not applicable).

Sample Number	D RH1	C RH2	E RH3	c RH4	e RH5	Reticulocyte Sorting Method	Number of Columns	*RHCE* Haplotypes
1	−	−	−	+	+	Size/Density	NA	*RHCE*ce (RHCE*01)/RHCE*ce (RHCE*01)*
2	+	+	−	+	+			*RHCE*ce (RHCE*01)/RHCE*Ce (RHCE*02)*
3	+	−	−	+	+	Antigen specificity	1	*RHCE*ce (RHCE*01)/RHCE*ce (RHCE*01)*
4	+	−	−	+	+			*RHCE*ce (RHCE*01)/RHCE*ce (RHCE*01)*
5	+	−	−	+	+			*RHCE*ce (RHCE*01)/RHCE*ce (RHCE*01)*
6	−	−	−	+	+		3	*RHCE*ce (RHCE*01)/RHCE*ce (RHCE*01)*
7	+	+	−	+	+			*RHCE*Ce (RHCE*02)/RHCE*ce (RHCE*01)*
8	+	−	+	+	+		4	*RHCE*ce (RHCE*01)/RHCE*cE (RHCE*03)*
9	+	+	−	−	+			*RHCE*Ce (RHCE*02)/RHCE*Ce (RHCE*02)*
10	+	−	−	+	+			*RHCE*ce (RHCE*01)/RHCE*ce (RHCE*01)*
11	+	+	+	+	+		6	*RHCE*Ce (RHCE*02)/RHCE*cE (RHCE*03)*
12	+	+	+	+	+			*RHCE*Ce (RHCE*02)/RHCE*cE (RHCE*03)*

**Table 2 ijms-25-05868-t002:** Polymorphisms discriminating between *RHD* and *RHCE* sequences (respective GenBank accession numbers NM_016124.6 and NM_020485.8) used to assess *RHCE*-specific RT-PCR.

Position (mRNA)	rs Number	Location	*RHCE*	*RHD*
c.361	rs1053345	Ex. 3	A	T
c.380	rs1053346	Ex. 3	C	T
c.383	rs1053347	Ex. 3	G	A
c.455	rs35109888	Ex. 3	C	A
c.505	rs1020280601	Ex. 4	C	A
c.509	rs987753117	Ex. 4	G	T
c.514	rs1053349	Ex. 4	T	A
c.544	rs1053350	Ex. 4	A	T
c.577	rs1384157219	Ex. 4	A	G
c.594	rs1053354	Ex. 4	T	A
c.602	rs141398055	Ex. 4	G	C
c.667	rs147357308	Ex. 5	G	T
c.697	rs142246017	Ex. 5	C	G
c.712	rs144163296	Ex. 5	A	G
c.733	rs1053361	Ex. 5	C	G
c.744	rs149352457	Ex. 5	T	C

**Table 3 ijms-25-05868-t003:** Polymorphisms discriminating between *RHCE**c and *RHCE***C* alleles [11].

Position (mRNA)	rs Number	Location	*RHCE**c	*RHCE***C*
c.048	rs586178	Ex. 1	G	C
c.150	rs200955066	Ex. 2	C	T
c.178	rs181860403	Ex. 2	C	A
c.201	rs1053343	Ex. 2	A	G
c.203	rs1053344	Ex. 2	A	G
c.307	rs676785	Ex. 2	C	T

## Data Availability

The data that support the findings of this study are available from the corresponding author, J.D.C., upon reasonable request.

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
