# Peer review of "Resolution of RHCE Haplotype Ambiguities in Transfusion Settings"

_ijms, 2024, doi:10.3390/ijms25115868_

Round 1

Reviewer 1 Report

Comments and Suggestions for Authors

The Authors are to be given credit for extending the previous work on RHCE alleles in a clinical setting. Indeed, it would have been appropriate to refer e.g. to Pedini et al (reference 31) in the Introduction in order to give a broader context to the work instead of only using it as a technical reference. On the other hand, the list of typing ambiquities on the rows 68-72 lacks a reference which would be useful.

Some of the statements given by the Authors are slightly exaggerating like row 28 saying that transfusion would be "the main therapeutic option" for autoimmune hemolytic anemia or stem cell transplantation. Certainly transfusion is important in these therapies but hardly the main option.

The Authors are to be congratulated for their success using CD71 enrichment for reticulocytes to facilitate RNA extraction; this has not been the case for some other research groups. It would be interesting to know if the Authors have made any further molecular controls on the level of leukocyte/platelet RNA contamination on their reticulocyte RNA like analysing leukocyte specific transcripts? I did not find in the text any information on the basic spectrophotometric quantification of RNA; maybe this is because of low yields?

Since the title of the manuscript (RHCE haplotype ambiguities disclosure in transfusion settings) hints to clinical application it is somewhat disturbing that one of the case studies briefly presented did not lead to unambigous result.  Since the techiques described represent very advanced level testing for transfusion compatibility, it would have been useful to open slightly more the transfusion phenotype recommendation given in this case.

The relationship between RHD and RHCE has only briefly been mentioned. I believe that in some situations the results on the RHD and RHCE analyses need to be combined. This aspect could have been mentioned in the Discussion.

Comments on the Quality of English Language

The language of the Authors is understandable and logical. The technical nature of the manuscript means that it cannot be very easy to read with numerous phenotype, genotype and haplotype notations punctuating the flow of the text. The Authors have done an appropriate effort to overcome this. 

Sequences of words and formulations in some instances sound slightly unnatural and using a native scientific English consultant might polish the text. Some sentences are too complicated; to give an example: "In immuno-hematology routine, serological techniques drawbacks are supplemented by genotyping methods [12-18], however, some samples happen to show ambiguities or discrepancies that require custom analyses.". This sentence on immunohematology would benefit from splitting to two.

Author Response

The Authors are to be given credit for extending the previous work on RHCE alleles in a clinical setting. Indeed, it would have been appropriate to refer e.g. to Pedini et al (reference 31) in the Introduction in order to give a broader context to the work instead of only using it as a technical reference.

The authors thank the reviewer; previous work conducted by French Blood Center research teams on RHCE alleles in clinical setting have been added in the introduction as follows: “Still, in some situations, ambiguities or discrepancies cannot be resolved, especially for RhCE, and uncertainties remain as for transfusion instructions; in this context, the French Blood Center, whose main mission is to ensure France's self-sufficiency in blood products under optimal safety and quality conditions, is continuously setting up molecular typing strategies in line with technological developments (18-29) ».

On the other hand, the list of typing ambiquities on the rows 68-72 lacks a reference which would be useful.

The authors agree with the reviewer, although the typing ambiquities have been listed from our experiences, references have been added in the introduction: “From our experience, the most reported RhCE typing ambiguities in clinical context are ceSL/CE [RHCE*01.10/*04] or CeJAHK/cE [RHCE*02.03/*03]; Ce(667)/ce(W16C) [RHCE*02.22/*01.01] or Ce/ceMO [RHCE*02/01.07]; CE/ceMo [RHCE*04/01.07] or Ce(667)/cE [RHCE*02.22/*03]; ceHAR/Ce [RHCE*01.22/*02] or CeVA/ce [RHCE*02.04/*01] and ceMo/CE [RHCE*01.07/04] or Ce(667)/cE [RHCE*02.22/*03] (4, 11, 30-32).”

Some of the statements given by the Authors are slightly exaggerating like row 28 saying that transfusion would be "the main therapeutic option" for autoimmune hemolytic anemia or stem cell transplantation. Certainly transfusion is important in these therapies but hardly the main option.

The authors thank the reviewer; this affirmation has been modified :” Red blood cell (RBC) transfusion is an important therapeutic option”.

The Authors are to be congratulated for their success using CD71 enrichment for reticulocytes to facilitate RNA extraction; this has not been the case for some other research groups. It would be interesting to know if the Authors have made any further molecular controls on the level of leukocyte/platelet RNA contamination on their reticulocyte RNA like analysing leukocyte specific transcripts? I did not find in the text any information on the basic spectrophotometric quantification of RNA; maybe this is because of low yields?

The authors thank the reviewer; no analysis was performed to check for leukocyte or platelet RNA contamination, this was added in the discussion as follows: “No cytometry or molecular biology analysis was carried out to check for leukocyte or platelet contamination.”  RNA was quantified with nanodrop and 100 ng was used for reverse transcription (see M&M section). No issue was encountered with low yield of RNA with the reticulocytes antigen-specific isolation method. We feel like the RNA extraction with Trizol is a performant method to obtain good quantity and quality of nucleic acid.

Since the title of the manuscript (RHCE haplotype ambiguities disclosure in transfusion settings) hints to clinical application it is somewhat disturbing that one of the case studies briefly presented did not lead to unambigous result. 

The authors thank the reviewers. Both cases presented were successfully resolved. Modification was made for more clarity in the results section: “The first case was resolved and displayed a RHCE*02/RHCE*03.13 genotype, i.e., displaying a c.728A>G polymorphism in the 5th exon associated to the RHCE*cE allele (RHCE*03.13) (Figure S5). Therefore, the phenotype was confirmed as C, Ew, cw, e, transfusion order was RH:-3, -4.”

Since the techiques described represent very advanced level testing for transfusion compatibility, it would have been useful to open slightly more the transfusion phenotype recommendation given in this case.

The authors agree with the reviewer, transfusion order for the first case was added in the results section: “Therefore, the phenotype was confirmed as C, Ew, cw, e, transfusion order was RH:-3, -4.” and “considering their C/e phenotype, transfusion order was RH:-3, -4. “

The relationship between RHD and RHCE has only briefly been mentioned. I believe that in some situations the results on the RHD and RHCE analyses need to be combined. This aspect could have been mentioned in the Discussion.

The authors agree with the reviewer, a paragraph at the end of the discussion has been added accordingly: “It should be stressed that analysis of the Rh system requires a deep knowledge of the existing variants and the relationship between the homologous RHD and RHCE genes, whatever the methodological developments and progress made in that field. In some situations, Rh phenotype identification requires analysis of both genes; for example a Cw phenotype, in absence of the 109pb insertion in exon 2 of RHCE, may be due to an RHD*DAU5 allele (19); an RhD phenotype, in absence of the RHD gene, may be due to an RHCE*ceHAR allele (44); identification of an RH:-18 phenotype in an RHD*DAR context requires definition of the RHCE genotype, i.e. RHCE*ceAR/RHCE*ce or RHCE*ceAR/RHCE*ceAR.

In conclusion, accurate identification of RhD and RhCE phenotype remains a complex topic that calls for complementary methodologies, expertise in genetics and experience in immunohematology.”

The language of the Authors is understandable and logical. The technical nature of the manuscript means that it cannot be very easy to read with numerous phenotype, genotype and haplotype notations punctuating the flow of the text. The Authors have done an appropriate effort to overcome this. 

The authors thank the reviewer.

Sequences of words and formulations in some instances sound slightly unnatural and using a native scientific English consultant might polish the text. Some sentences are too complicated; to give an example: "In immuno-hematology routine, serological techniques drawbacks are supplemented by genotyping methods [12-18], however, some samples happen to show ambiguities or discrepancies that require custom analyses.". This sentence on immunohematology would benefit from splitting to two.

The authors agree with the reviewer, English has been revised.

Reviewer 2 Report

Comments and Suggestions for Authors

In the paper of Izard and colleagues authors tested two protocols for reticulocyte isolation and proved that sorting method by antigen-specificity from EDTA blood samples collected up-to 48 hours is the more efficient; and that the association of a RHCE-specific RT-PCR followed by a RHCE allele-specific sequencing allows cDNA RHCE haplo-types analysis. The paper is well organized and the results add important information to the field. 

However I have to raise some following comments: 

1. On the Figure 1 "a" is ACTB (actinβ) mRNA and "b" is 18S ribosomal RNA real-time PCR analysis. However, in the legend to Figure 1 "a" is indicated as 18S ribosomal RNA and "b" is indicted as ACTB (actinβ) mRNA.

2. On the page 3, lines 106-107, authors cliamed that "the use of three columns displays better yield as compared to one column, four or six columns per sample (Figure 1)". How authors came to such conclusion considering results presented on the Figure 1?

3. On the page 3, line 118: I suggest to use more than twelve samples for validation protocol.

4. Table 1: Authors should explain in the table legend or below the table what are the columns with D, C, c, E, e.

5. Table 2 and 5 should be considered in the Supplementary material

Comments on the Quality of English Language

Minor editing of English language required

Author Response

  1. On the Figure 1 "a" is ACTB (actinβ) mRNA and "b" is 18S ribosomal RNA real-time PCR analysis. However, in the legend to Figure 1 "a" is indicated as 18S ribosomal RNA and "b" is indicted as ACTB (actinβ) mRNA.

The authors thank the reviewer; the error has been corrected.

  1. On the page 3, lines 106-107, authors cliamed that "the use of three columns displays better yield as compared to one column, four or six columns per sample (Figure 1)". How authors came to such conclusion considering results presented on the Figure 1?

The authors agree with the reviewer; more explanation was given as follows in the results section “Indeed, no difference was observed in 18S ribosomal RNA real-time PCR results between three, four or six columns and the control.”

  1. On the page 3, line 118: I suggest to use more than twelve samples for validation protocol.

The authors thank the reviewer, the protocole of RHCE-specific RT-PCR and haplotype specific-sequencing for RHCE haplotypes exploration has been validated on more RNA samples extracted from paxgen RNA tube, excluding the reticulocyte isolation method. We initially thought that the inclusion of this part would have burden the article, however, we agree with the reviewer and included this validation step. In M&M section: “RNA isolated from peripheral blood collected in PAXgene® Blood RNA tubes from Sickle Cell Disease patients enrolled at the Internal Medicine Department, AP-HM, Marseille after obtaining written informed consent (protocol approved by the Ethics Committee Sud Mediterranée IV; ID-RCB: 2017-A02744-49) with known RHCE genotype and RhCE phenotype was used as controls in molecular biology experiments. “ and in results section: “The analysis of the 30 RNA samples extracted from peripheral blood collected in PAXgene® Blood RNA tubes used to validate the RHCE mRNA protocol also revealed a full concordance of RhCE phenotype and RHCE haplotype sequencing (Table S3). “

  1. Table 1: Authors should explain in the table legend or below the table what are the columns with D, C, c, E, e.

The authors thank the reviewer; the table legend has been revised.

  1. Table 2 and 5 should be considered in the Supplementary material

The authors agree with the reviewer; Table 2 and 5 have been placed in the Supplementary material, table numeration has been modified accordingly.